# A Sample-Based Method for Semantic Understanding of Neural Network Decisions

## Abstract

Interpretability in deep learning is one of the largest obstacles to its more widespread adoption in critical applications. A variety of methods have been introduced to understand and explain decisions made by Deep Models. A class of these methods highlights which features are most influential to model predictions. These methods have some key weaknesses. First, most of these methods are applicable only to the atomic elements that make up raw inputs to the model (e.g. pixels or words). Second, these methods generally do not distinguish between the importance of features individually and their importance due to interactions with other features. As a result, it is difficult to explore high-level questions about how models use features during decision-making. We tackle these issues by proposing Sample-Based Semantic Analysis (SBSA). We use Sobol variance decomposition as our sample-based method which allows us to quantify the importance of semantic combinations of raw inputs and highlight the extent to which these features are important individually as opposed to due to interactions with other features. We demonstrate the ability of Sobol-SBSA to answer a richer class of questions about the behavior of Deep Learning models by exploring how CNN models from AlexNet to DenseNet use regions when classifying images. We present three key findings. 1) The architectural improvements from AlexNet to DenseNet manifested themselves in CNN models utilizing greater levels of region interactions for predictions. 2) These same architectural improvements increased the importance that CNN models placed on the background of images 3) Adversarially robust CNNs reduce the reliance of modern CNNs on both interactions and image background. Our proposed method is generalizable to a wide variety of network and input types and can help provide greater clarity about model decisions.

## 1 Introduction

Deep learning models are becoming endemic in various applications. As models are increasingly used for critical applications in medicine such as detecting lung nodules (Schultheiss et al., 2021) or autonomous driving (Li et al., 2021), it is important to either create interpretable models or to make opaque models human interpretable. This paper focuses on the latter. Existing methods developed over the last decade for doing this can be broken down into model agnostic vs model dependent. Model agnostic methods, such as Shapley values (Kononenko et al., 2013) and Integrated Gradients (Sundararajan et al., 2017) weigh the importance of input features without relying on the structure of the model. In contrast, methods such as GradCam (Selvaraju et al., 2017) and GradCam++ (Chattopadhay et al., 2018) are heavily dependent on model architecture.

While these methods yield valuable information about models, they share common gaps. First, they do not distinguish between the features in input space that are individually important and features that are important because of their interaction with other features. Second, the above methods are generally applied to inputs at their most granular level (pixels, words, etc..). The combination of these gaps limits the conclusions that Machine Learning practitioners can make about the behavior of models as a whole.

We address these limitations in two key ways. First, we introduce a two-part framework called Sample-Based Semantic Analysis (SBSA). The first part of the framework is a function that generates semantic representations of inputs and associates these semantic representations with real

numbers. The second part of the framework is a black-box sample-based sensitivity method. In this case, the Sobol method which reports the importance of individual features and their interactions. Second, we demonstrate the ability of Sobol-SBSA to answer a richer set of questions than standard interpretability methods by applying it to CNN models in the context of ImageNet.

The key results and contributions of this paper are as follows:

1. We present a general-purpose framework for using sample-based sensitivity methods to analyze the importance of semantic representations of inputs and test it using a variety of black-box methods.

2. We demonstrate that the Sobol method outperforms other popular black box methods, Integrated Gradients, Shapley (Kernel Shap), and LIME, for selecting both the most and least important regions to CNN predictions.

3. We show, *through direct measurement*, that the main impacts of the evolution of CNN architectures were increasing the extent to which they used region interactions and by which they relied on background information in images. Similarly, we show that adversarially robust versions of CNNs reduce both of these effects for modern CNNs. To our knowledge, *Sobol-SBSA is the first pipeline to facilitate the direct measurement of such trends, and to do so within a single pipeline.*

## 2 METHODOLOGY

In this section, we describe the two components of SBSA and specify how we use it to analyze the importance of image regions in ImageNet. In particular, we describe how we associate image regions to quantities that can be analyzed with a sampling-based method, and the specifics of Sobol as a sampling-based sensitivity method.

### 2.1 SAMPLE-BASED SEMANTIC ANALYSIS (SBSA)

Let us define the following variables. $x \in \mathbf{R}^d$ is an input to a model, $f : x \to y \in \mathbf{R}^s$ is a model that takes $x$ as an argument and produces $y$, $\tilde{x}^{[i]} \in \mathbf{R}^d$ is a sample of $x$, and $N \in \mathbf{Z}$ is a prescribed integer that helps to determine the number of $\tilde{x}^{[i]}$ samples generated. Most sample-based sensitivity methods operate by generating a number of samples that is some function of $N$ and $d$. The model is then evaluated on these samples and the resulting model outputs are used by Sensitivity analysis methods, such as Sobol, to determine the importance of components of $x$ to the model output, $y$.

One thing that immediately becomes clear is that for deep learning applications with high-dimensional inputs, such as images, videos, and long documents, applying this process naively is prohibitively expensive. This issue can be greatly minimized if one turns to semantic representations of inputs instead. In this paper, a semantic representation of an input is defined as follows. **A semantic representation of an input, $x$, is some combination of the raw components of that input which yields a human recognizable higher order feature, such as the colors in an image, image regions, or grammatical parts of sentences**. We define this semantic representation as $\{S_1, \ldots S_l\}, S_k \in \mathbf{R}^m$, where $m < d$. Recalling that most sample-based sensitivity methods operate on real numbers, we define three mapping objects.

$$G : x \to \{S_1, \ldots, S_l\},\ G^{-1} : \{S_1, \ldots, S_l\} \to \approx x \quad x \in \mathbf{R}^d,\ S_k \in \mathbf{R}^m,\ l < d,\ m < d \tag{1}$$

$$H : \{S_1, \ldots, S_l\}, \to \{r_1, \ldots, r_l\}, \quad S_k \in \mathbf{R}^m,\ r_k \in \mathbf{R}, \tag{2}$$

$$R : \{(r_1, S_1), \ldots, (r_l, S_l)) \to \{S_1^*, \ldots, S_l^*\}, \quad S_k^* \in \mathbf{R}^m \tag{3}$$

$G$ maps the raw input, $x$, to $l$ semantic representations, $S_k$, $H$ associates the semantic representation to some lower dimension vector of real numbers, $r \in \mathbf{R}^l$, and $R$ creates new semantic representations based on $r_k$ and $S_k$. $G$ is invertible. SBSA generates samples of $r$, $[\tilde{r}^{[1]}, \ldots, \tilde{r}^{[n]}]$. From these samples, $R$ is used to generate samples of the original semantic representations, $R\big(\tilde{r}_k^{[i]}, S_k\big) = \tilde{S}_k^{[i]}$,

and $G^{-1}$ uses these samples to create samples of the raw inputs, $\tilde{x}^{[i]}$.

$$\{R(\tilde{r}_1^{[i]}, S_1), \ldots, R(\tilde{r}_l^{[n]}, S_l)\} = \{\tilde{S}_1^{[i]}, \ldots, \tilde{S}_l^{[n]}\} \tag{4}$$

$$G^{-1}(\tilde{S}_1^{[i]}, \ldots, \tilde{S}_l^{[i]}) = \tilde{x}^{[i]} \tag{5}$$

The model is then evaluated on the samples of the raw input, $\tilde{x}^{[i]}$. Since the sampling was done in $r$, the sensitivity analysis reports the importance of components of the semantic representations of the input, $S_k$ corresponding to the components of $r$, $r_k$. $G$, $H$, and $R$ can be any functions that derive $S_k$ from $x$, associates a real number, $r_k$, with $S_k$, and produces a semantic representation, $S_k^*$, as a function of $r_k$ and $S_k$. However, we recommend two properties to maximize our approach. **Property 1 (Sensitivity)** *Any change in $r$ should result in a change $x$.* Sample-based sensitivity methods observe how changing different input components impact model outputs in order to determine importance. **Property 1** insures that all samples in $r$ contribute information to SBSA. **Property 2 (Approximate Reconstruction)**: *When using $S$ and $r$ produced from the original input, $R$ and $G^{-1}$ should closely approximate $x$.* Sample-based sensitivity analysis methods construct samples that are uniformly distributed between 0 and 1, and scales these samples to bounds determined by the original $r$ (Herman & Usher, 2017). **Property 2** insures that produced samples of $x$, $\tilde{x}^{[i]}$, will increase and decrease different semantic components of $x$ with approximately equal probability.

In the following section we describe the use of SBSA via image regions.

## 2.2 Sample-Based Semantic Analysis (SBSA) Applied to ImageNet and Regions

In this section, we discuss how we apply our pipeline to the scenario in which the semantic features of interest are regions in an image, and the sample-based sensitivity method is Sobol (Sobol-SBSA). We first describe our mapping functions for regions, then we detail the Sobol method for sensitivity analysis and the relevant measures that it produces.

Given an image $x \in \mathbf{R}^d$ where $d = W \times L$, our input to semantic features function, $G$, extracts $l$ regions and normalizes them by the sum of pixels in those regions. The mapping, $H$, associates each region, $S_k$, with the sum of pixels in that region, $r_k$. These values are used to construct the vector $r \in \mathbf{R}^l$.

$$G : x \in \mathbf{R}^d \to \{S_1, \ldots, S_l\}, \quad S_k = \frac{x_{\{t,p\} \in S_k}}{\sum_{\{t,p\} \in S_k} x_{tp}} \tag{6}$$

$$H : \{S_1, \ldots, S_l\} \to \{r_1, \ldots, r_l\}, \quad r_j = \sum_{\{t,p\} \in S_k} x_{tp} \tag{7}$$

The function, $R$, multiplies each semantic feature, $S_k$, with the associated value, $r_k$ and finally $G^{-1}$ creates $x$ by stitching together the regions to reform $x$. Mathematically, this is as follows.

$$R : \{(r_1, S_1), \ldots, (r_l, S_l)\} \to \{S_1^*, \ldots, S_l^*\}, \quad S_k^* = r_k S_k \tag{8}$$

$$G^{-1} : \{S_1, \ldots, S_l\} \to x \tag{9}$$

The above mapping, while simple, satisfies the properties of sensitivity and approximate reconstruction. As a result the samples of $r$, $\tilde{r}^{[i]}$, produce image samples, $\tilde{x}^{[i]}$, that amplify and mask regions of the image with relatively equal probability and uniformly. The Saltelli method, (Saltelli et al., 2010), which chooses optimal points in $[0, 1]$ for the Sobol method, are used to construct these samples and the model outputs of these image samples are fed to the Sobol method.

We will briefly give an overview of the Sobol method. For more details about the implementation of Sobol, and the associated sampling method, see Saltelli et al. (2010) and Herman & Usher (2017). The Sobol method is a variance-based sensitivity method. Given a model, $f : (x_1, \ldots, x_d) \in \mathbf{R}^d \to y \in \mathbf{R}^s$, the variance based first order effect of a component of the input, $x$, is:

$$V_i = V_{x_i}(E_{\mathbf{x} \sim i}(y_j | x_i)) \tag{10}$$

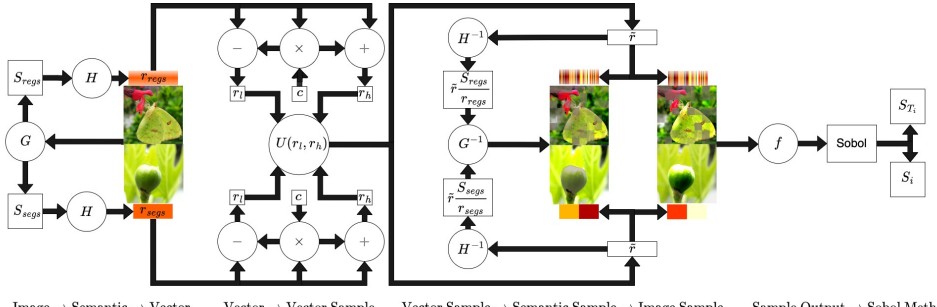

Image → Semantic → Vector     Vector → Vector Sample     Vector Sample → Semantic Sample → Image Sample     Sample Output → Sobol Method

Figure 1: Sobol-SBSA pipeline for regions and segments (ImageNet-S). **1) Image → Semantic → Vector** extracts semantic features from the image and maps these onto real numbers. **2) Vector → Vector Sample** produces samples from the vector. **3) Vector Sample → Semantic Sample → Image Sample** produces samples of the image that mask and amplify semantic features based on the vector samples. **4) Sample Output → Sobol Method** sends outputs from the model evaluated on the image samples to the Sobol method for analysis.

where $y_j$ is a component of the model output, $x_i$ is a component of the input, $x$, $\mathbf{x}_{\sim i}$ are samples of the input where all components of the input except for $i$ are varied, and $V_{x_i}$ and $E_{\mathbf{x}\sim i}$ are the variance and expectation over $x_i$ and $\mathbf{x}_{\sim i}$ respectively. The Sobol first-order sensitivity is then the ratio of $V_i$ to the variance of the model output, $y_j$.

$$S_i = \frac{V_i}{V(y_j)} \tag{11}$$

$S_i$ quantifies the individual importance of the $i^{th}$ component of the input, $x$, to the $j^{th}$ component of the output, $y$. This equation essentially states that Sobol first order index, $S_i$, is the fraction of the variance of the output, $y_j$ that is accounted for by $x_i$. What makes Sobol unique is that it simultaneously calculates $S_i$ and $S_{T_i}$, the total effect index. The total effect index, $S_{T_i}$, is the importance of feature $i$ due to both the feature independently and every higher level interaction of this feature with other features.

$$S_{T_i} = 1 - \frac{V_{\mathbf{x}\sim i}(E_{x_i}(y_j|\mathbf{x}_{\sim i}))}{V(y_j)} = S_i + \sum_{k;k\neq i} S_{ik} + \sum_{k,j;k\neq i,j;j\neq i} S_{ikj} + \ldots \tag{12}$$

Dividing both sides of equation 12 by $S_{Ti}$ yields

$$1 = \frac{S_i}{S_{Ti}} + \sum_{k;k\neq i} \frac{S_{ik}}{S_{Ti}} + \sum_{k,j;k\neq i,j;j\neq i} \frac{S_{ikj}}{S_{Ti}} + \cdots = PIR + \ldots \tag{13}$$

The first term in equation 13 reports the extent to which a feature, $x_i$, is important to the model output due to the feature by itself. The larger the term, the greater the importance of the region by itself, contrarily, the smaller the term larger the importance of the interaction of the feature with other features. We will refer to this as the Primary Index Ratio ($PIR$) for the rest of the paper. We note that the number of samples used for the Sobol method is $N(d+2)$ (Saltelli et al., 2010). For all of the experiments that follow, $N = 50$ and $d = l$, the number of semantic features.

### 2.2.1 Choice of Semantic Features

In order to demonstrate the ability of our method to address high level questions about CNNs at a semantic level, we apply our pipelines to three types of regions. 1) Equally sized image patches 2) machine annotated segmentations of the ImageNet training set from Salient ImageNet Singla & Feizi (2021) and 3) human annotated segmentations from the ImageNet-S 919 validation set (Gao et al., 2021). Salient ImageNet segments images into core regions (regions that should be important for predictions) and spurious ones. ImageNet-S is a dataset that was created for use in judging image segmentation task. *A key difference between ImageNet-S and the other types of regions is*

*that ImageNet-S strictly respects boundaries between objects and background in an image. Region patches and Salient ImageNet do not.* Figure 1 shows our pipeline when applied to evenly size regions of an image and to ImageNet-S 919 segments.

## 3 EXPERIMENTS

### 3.1 VALIDATING OUR PIPELINE

We validate our pipeline in two ways. First, we demonstrate it's ability to accurately rank the importance of regions of different sizes. Second, we demonstrate that *Sobol-SBSA accurately reproduces two trends in how CNNs use regions that were determined indirectly by two mutually exclusive papers.*

- Brendel & Bethge (2019) showed that more modern CNNs use higher levels of interactions between equally sized region patches than older models.
- Singla et al. (2022) showing that robust versions of CNN models utilize "spurious" areas of an image, as determined by machine annotation, less than normal versions of these CNNs.

Having established trust in Sobol-SBSA, we demonstrate how it can be used to further explore how CNNs use regions, without needing to use a different pipeline. First, we demonstrate the impact of semantic representation on Brendel & Bethge (2019)'s results by measuring the interaction between segmented objects and their backgrounds (ImageNet-S). We show that *the trend of more modern CNNs utilizing interactions more than older models is only clear when viewed with respect to regions that do not strictly respect object boundaries.* Second, we demonstrate that, regardless of whether or not the segmentation respects boundaries, *a result of the development of CNN architectures was greater exploitation of background information when making decisions.* Finally, we demonstrate that, *robust models reduce the extent to which models use the interaction if regions which do not respect boundaries.* To the authors' knowledge, this is the first time that all of these results have been demonstrated through direct measurement and in a single pipeline. **We note that all ImageNet models were pre-trained Pytorch models from ImageNet1K_V1.**

### 3.2 VALIDATING SOBOL-SBSA FOR IMPORTANCE RANKING AND SEMANTIC UNDERSTANDING

Our SBSA model can be used with a variety of black box models. Thus, we compare the following two methods against Sobol-SBSA.

- **Shapley (Shap-SBSA) (Kononenko et al., 2013):** Used with SBSA in a similar manner as Sobol-SBSA.
- **LIME (LIME-SBSA) (Ribeiro et al., 2016):** Used with SBSA in a similar manner as Sobol-SBSA.

We also compare against the following:

- **Integrated Gradients (Sundararajan et al., 2017):** Cannot be used with SBSA since it requires gradients with respect to pixels. Instead, we aggregate pixel importance in regions and rank region importance based on this aggregation.
- **Random:** A control baseline where regions are randomly selected.
- **Ideal:** The ideal result based on our metric. This is not experimental.

We use two standard metrics for comparing interpretability methods. First, we measure the change in the ground truth label score of a model when regions, as specified by the sensitivity method, are masked. We mask the top and bottom 20% of regions in an image. Second, we measure sensitivity-$n$ correlation. Sensitivity-$n$ correlation is a quantity proposed by Ancona et al. (2017) for comparing the effectiveness of attribution methods when determining which features are important to a model. Quantitatively, it is a measurement of the Pearson correlation between $\sum_{i=1}^{n} R_i^c$ and $S_c(x) - S_c(x_{[x_S=0]})$. $\sum_{i=1}^{n} R_i^c$ is the sum of the attributions associated with the $n$ input features that

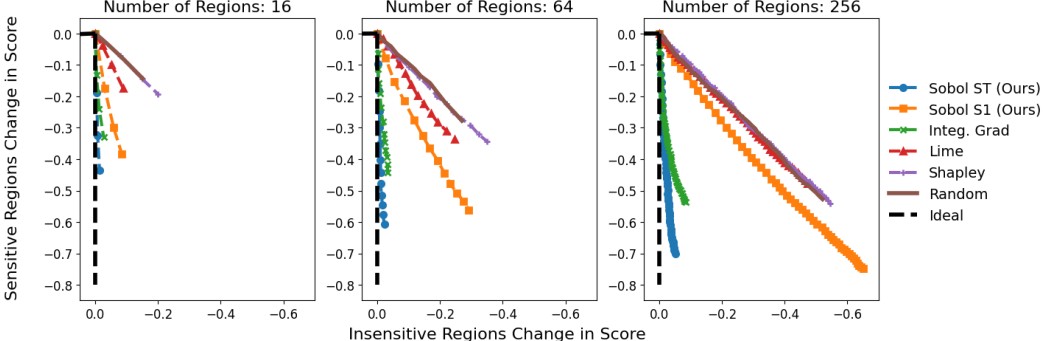

Figure 2: Plots of the change in the ground truth score of the ResNet50 when the top 20% of regions are masked vs the bottom 20% of regions. A steep slope is indicative of a more effective method since it means that the change in ground truth scores are significant when the top 20% of images are masked, but not when the bottom 20% of images are masked. Our measure, $S_{T_i}$ is closest to the ideal.

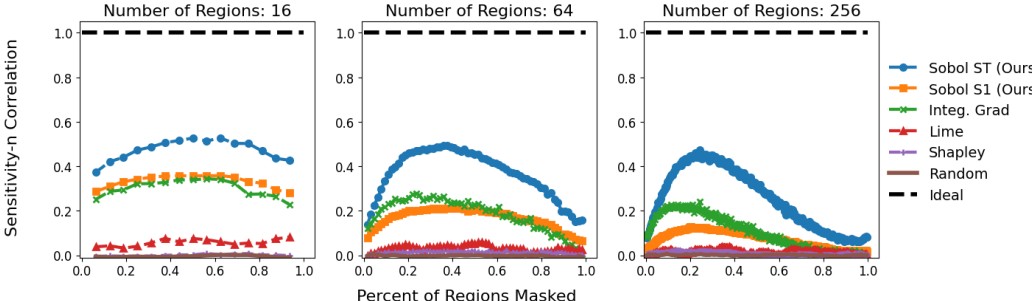

Figure 3: Plots of the sensitivity-$n$ correlation for different sensitivity methods as a function of the percentage of regions masked. The sensitivity methods with values closer to one are more effective. Our measure, $S_{T_i}$ is closest to the ideal.

are masked and $S_c(x) - S_c(x_{[x_S=0]})$ is the difference in the score that the model produces when $n$ input features are masked versus when none are. For each value of $i$, $i$ random features are selected to be masked 100 times and the correlation is averaged over the examined data. A value closer to one means a more effective method.

For all methods, we evaluate the importance of regions to ResNet50 on 1000 randomly selected ImageNet validation images. To explore the importance of image size, we performed our experiments for $4 \times 4$, $8 \times 8$, and $16 \times 16$ regions. The number of samples is selected as a function of the number of regions, as detailed in section 2.2. For $4 \times 4$, $8 \times 8$, and $16 \times 16$ regions, $d = 16$, $d = 64$, and $d = 256$ respectively.

Figures 2 and 3 show the results of the masking and sensitivity-$n$ correlations respectively for all evenly sized region patches. Figure 2 plots the change in the model score when the most important regions are masked versus when the least. A steeper curve is more ideal since it means that the sensitivity method significantly decreases the model score when masking the most important regions, but has little impact when masking the least important regions. *The Sobol-SBSA is the most effective by both the masking and sensitivity-$n$ measures*. The key takeaways from the plot are 1) $S_{Ti}$ performs the best overall in picking the most and least important regions in an image, as well as more properly ranking the regions in between (as measured by sensitivity-$n$), and 2) $S_{Ti}$ is the most robust to changes in region size, followed closely by Integrated Gradients. The gap in performance between $S_{T_i}$ and $S_i$ shows the importance of accounting for region interaction when ranking importance.

We now use Sobol-SBSA to quantify how the importance of interactions between evenly sized regions to CNN model outputs evolved over time, and test whether or not this matches the trend found by Brendel & Bethge (2019). Figure 4a plots the average $PIR$ for 10000 validation images

when run on $4 \times 4$, $8 \times 8$, and $14 \times 14$ regions. A lower $PIR$ means greater region interaction importance. We see that that for all of these regions, interaction decreases from AlexNet and VGG16 to the more modern InceptionV3, ResNet50, and DenseNet161 architectures, an identical result to Brendel & Bethge (2019)'s. Because Pytorch's pre-trained InceptionV3 takes as inputs images of size $299 \times 299$ rather than $224 \times 224$, the number of regions used for InceptionV3 are $5 \times 5$, $11 \times 11$, and $18 \times 18$. Finally, we apply Sobol-SBSA to 10000 Salient ImageNet images whose regions have been labeled as "core" regions that should be important to model predictions, and "spurious" regions that should not (Singla & Feizi, 2021). Figure 4b plots the average difference between the total index score of core and spurious regions for the normal and adversarially robust versions of VGG16_BN, ResNet50, and DenseNet161. We use pre-trained robust weights from Salman et al. (2020) where the $l_2$ threat radius was 3. For Salient ImageNet, the difference in core and spurious importance is greater for robust models than their normal counterparts. This compliments Singla et al. (2022)'s findings that robust CNN models relied less on spurious areas of images than their normal counterparts.

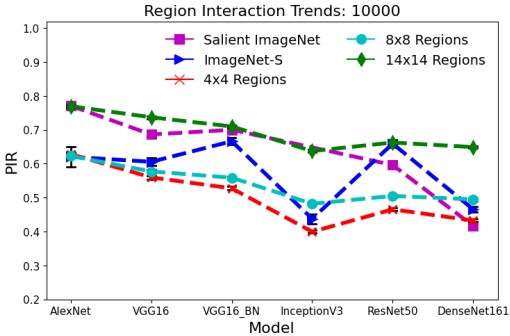 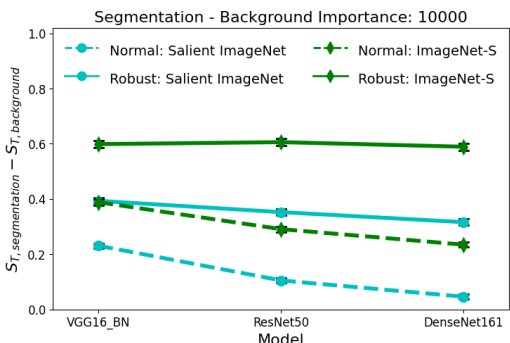

(a) The plots of the average mean of PIR for 10000 images for a series of CNN models when applied to images that are partitioned based on segmentations from ImageNet-S and Salient ImageNet, as well as when applied to the top 20% of region patches.

(b) The plots of the average difference in importance, $S_T$, for segmented objects in ImageNet-S and Salient ImageNet when applied to normal and robust models. 10000 images from each dataset were used.

Figure 4: PIR trends and the difference in importance between segmented objects and background for CNN models and their robust counterrparts.

### 3.3 BEYOND VALIDATION: THE IMPACT OF SEGMENTATION ON REGION IMPORTANCE AND INTERACTION TRENDS

In the previous subsection we validated Sobol-SBSA's ability to 1) correctly rank the importance of regions regardless of size 2) accurately capture trends in how CNNs use regions that were determined by Singla et al. (2022) and Brendel & Bethge (2019). We now explore the impact of segmentation type on the trends in region interaction and differences between robust and normal models. We also measure how the use of background information evolved with models.

Figure 4a plots the $PIR$ for Sobol-SBSA applied to foreground and background regions as determined by Salient ImageNet and ImageNet-S. For foreground and background areas of an image as determined by Salient ImageNet, interactions became more important with more modern CNNs. For ImageNet-S regions, however, these interactions only increased for DenseNet161 and InceptionV3 . Recalling that ImageNet-S regions are segments that strictly respect object boundaries, we conclude that while modern CNNs use greater interactions between regions that do not respect object boundaries, this is not consistently the case for those that do. InceptionV3 was not used with Salient ImageNet since the masks provided were $224 \times 224$ which is incompatible with the expected $299 \times 299$ input to InceptionV3. Figures 4b shows the average difference in the total index score, $S_{T_i}$, between foreground and background areas of images as determined by ImageNet-S and Salient ImageNet. This is done for normal and robust models. We see that the trend of robust models reducing the extent to which CNN architectures use background information holds, regardless of whether or not the foreground strictly respects object boundaries. Examples of this are shown in Figure 6. Figure 5a plots the average $PIR$ between foreground and background objects as deter-

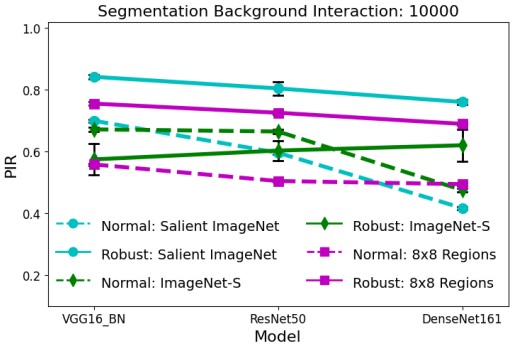 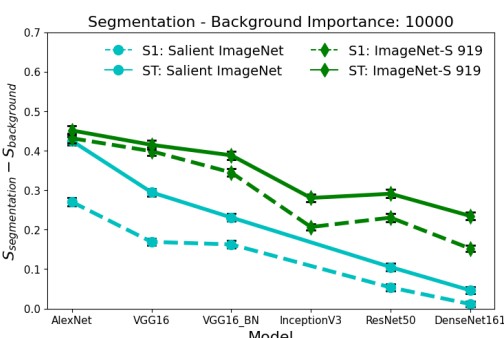

(a) The plots of the average mean of $PIR$ when Sobol-SBSA is applied to segmented and background objects in SalientImageNet and ImageNet-S for normal an robust models. 10000 images were used in their respective datasets.

(b) The plots of the average difference in importance, $S_T$ and $S_i$ for segmented objects in ImageNet-S and Salient ImageNet as a function of models. 10000 images from each dataset were used.

Figure 5: PIR trends and the difference in importance between segmented objects and background for CNN models

mined by ImageNet-S and Salient ImageNet for the robust and normal versions of CNNs. It is seen that, for areas defined by Salient ImageNet, robust models decreased the reliance of architectures on interactions, but that for objects and background determined ImageNet-S this was only clearly seen for DenseNet161. The key take away here is that robust models generally decrease model's reliance on interactions between regions that do not respect boundaries, but don't necessarily do so for those that do.

Finally, Figure 5b plots the average difference in importance between foreground and background objects when Sobol-SBSA is applied to ImageNet-S and Salient ImageNet. These results are plotted for AlexNet to DenseNet161. It is seen that one of the effects of architectural changes in CNN development was to increase the extent to which these models use background information.

### 3.3.1 THE LIMIT OF SOBOL-SBSA

While the Sobol-SBSA is a powerful method, it has a similar weakness to other sample-based black box methods. It assumes that the input features that it is analyzing are uncorrelated. When this is not the case, the ability of Sobol to decompose importance into independent subsets of the feature space is weakened (Li et al., 2010). This manifests itself in the Sobol Indices, which are supposed to satisfy the property that PIR $\leq 1$, having at least some input features for which PIR $> 1$.

The assumption that regions are uncorrelated breaks down as regions become smaller in size. We observe this in two ways. First, we examine figure 4a and note that $PIR$ increases as the region sizes decrease. This is intuitively an incorrect result since interactive effects should be greater for smaller regions, as demonstrated by Brendel & Bethge (2019). To confirm that this result is due to a breakdown in Sobol assumptions we calculate the average percent of regions for which PIR $> 1$ across all models for a given region size. For $4 \times 4$, $8 \times 8$, and $14 \times 14$ regions the percentages are 22%, 33%, and 39% respectively. Figure 8 shows an example of how Sobol-SBSA can breakdown with smaller region sizes. A key direction for future work is to implement a version of Sobol-SBSA that accounts for correlations in the data.

## 4 RELATED WORK

In recent years, interest has grown in interpretability methods that can quantify the importance of feature interactions. Janizek et al. (2020) introduced an extension to Integrated Gradients, called Integrated Hessians, that quantified the importance of pairwise interactions of features, while Zhang et al. (2020) built on this work by presenting a similar quantification through Shapley Interactions. While both methods are powerful, they require the separate calculation of individual feature interactions, as well as pairwise comparisons. Fel et al. (2021) used the Sobol method, but focused on

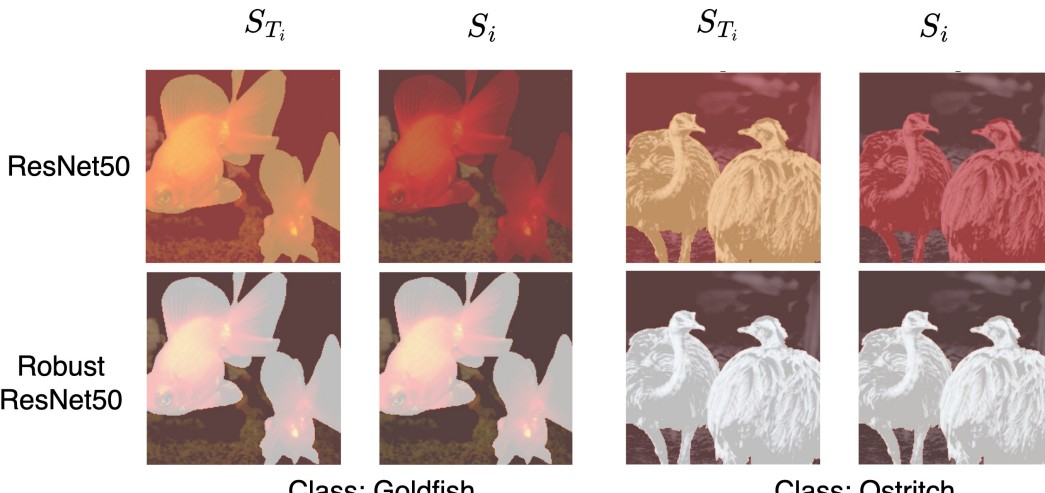

Figure 6: A sample result of the Sobol-SBSA when applied to an segmentations determined by the ImageNet-S dataset for the normal and robust versions of ResNet50. Robust ResNet50 focuses only on the segmented objects, while ResNet50 also uses the background.

importance, rather than interactions. None of the above works addressed understanding how models used semantic representations, or facilitated quantitatively answering high level questions about models.

## 5  DISCUSSION AND CONCLUSION

We proposed Sobol-SBSA, a general method for understanding how Deep Learning Models use semantic features when making decisions. We demonstrated the ability of this method to answer in one pipeline a rich set of questions about model behavior by using it to study how CNN models use areas of images during classification. We found 1) that the primary impact of the evolution of CNN models was to make greater use of region interactions and to increase the importance of image background to model predictions and 2) that adversarially robust CNN models are less susceptible to spurious correlations in the data because they force CNN architectures to rely less on region interactions and on image backgrounds.

The Sobol-SBSA method has a variety of potential applications beyond the image/region-based analysis that we presented here. Many different types of input partitioning and input modalities can be analyzed using the SBSA method. For images, the partitioning into image regions can be done via external metadata, such as object detection results. For natural language processing (NLP), the use of SBSA is even more straightforward since words/tokens provide a natural way to partition the input sequence. Hence, our model can be utilized to understand how parts of speech are utilized for different iterations of various NLP models, including translation, classification, and generative text models. Beyond unimodal content, SBSA can also be easily extended to multimodal setups. For example, Sobol-SBSA can be used to measure the strength of bias in Image Captioning systems by quantifying the extent to which parts of an image correspond to which parts of the generated caption. In addition to these applications, there are multiple avenues for future work. These include exploring the combination of Sobol-SBSA with automatic feature detectors, implementing a version of Sobol-SBSA that accounts for input correlations, and exploring whether $PIR$ can be used as a proxy for the robustness of non-CNN Deep Learning models.

Through SBSA and Sobol-SBSA we have proposed a strong foundation for obtaining a richer understanding of how models use semantic representations of inputs, regardless of whether these representations are generated automatically or by end users. We hope that it can be used to both provide clarity about the mechanisms by which Deep Learning makes decisions, and to influence how we further develop these models.

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

# A APPENDIX

## A.1 10K VS 50K EXPERIMENTS

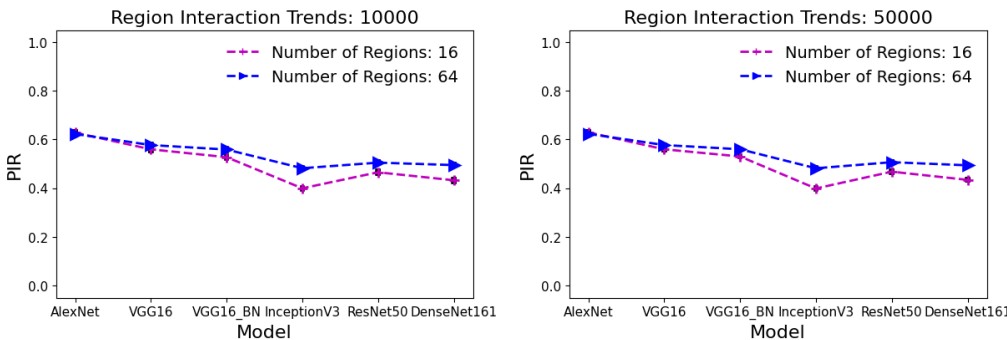

Figure 7: Average PIR for the top 20% of regions when calculated for 10000 and 50000 ImageNet validation images. This is done for $4 \times 4$ and $8 \times 8$ regions. The results are identical.

We calculated the average PIR for the top 20% of regions for 10000 and 50000 validation images in ImageNet using $4 \times 4$ and $8 \times 8$ regions respectively. We saw that the results were identical so, for the smaller regions generated when splitting an image into $14 \times 14$ regions, we calculated the average PIR for 10000 images to save computational cost.

## A.2 IMPACT OF REGION SIZE

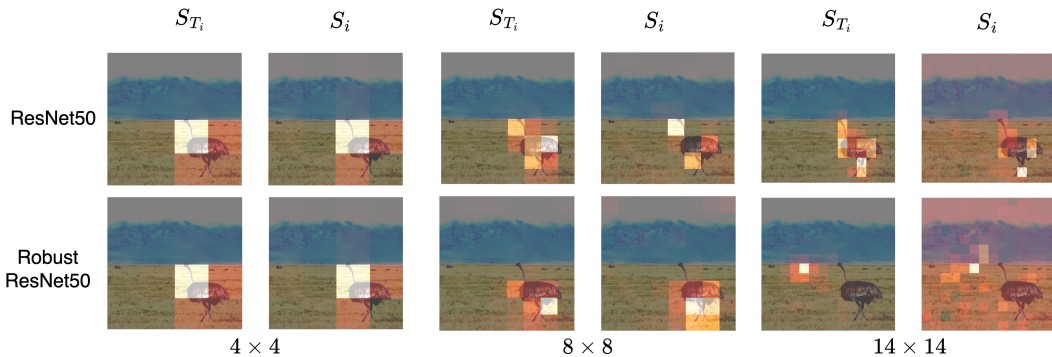

Figure 8: Images of $S_i$ and $S_{T_i}$ for an Ostrich class example for different grid sizes. We see that $4 \times 4$ and $8 \times 8$ regions are consistent, but that this is not the case for $14 \times 14$ regions.

Figure 8 shows an example of Sobol-SBSA when applied to an Ostrich target class for different region sizes. We see that all of the examples are consistent except for the Robust ReseNet50 $14 \times 14$ example. One of the weaknesses of Sobol is that the results can be corrupted when the input features are correlated. As regions get smaller, this is exactly what occurs. Although, overall, Sobol-SBSA was still able to correctly identify the most and least important regions at the $14 \times 14$ scale, figure 2, more work must be done to address this weakness so that the Sobol results can be compared accurately across different sizes of regions or, more generally, features that have different types of correlations. Future work will involve exploring the impact of sampling size and variations of Sobol that account of input correlation on addressing this issue.

