# OpenReview forum: "A Sample Based Method for Understanding The Decisions of Neural Networks Semantically"
_ICLR.cc/2023/Conference — Submitted to ICLR 2023_

### Official Review · Reviewer_phNs · 2022-10-25

**Confidence:** 5
**Correctness:** 3
**Technical Novelty And Significance:** 2
**Empirical Novelty And Significance:** 2
**Recommendation:** 5

**Clarity, Quality, Novelty And Reproducibility:**

The paper is clearly written, the novelty is incremental, and the approach can be reproduced.

**Strength And Weaknesses:**

Strengths:

1. The proposed sample-based approach to interpreting the decision-making of neural networks is interesting.

2. The Sobol variance-based sensitivity metric is a simple yet effective approach to estimating the importance of regions.

3. Authors conducted extensive experiments to evaluate various aspects of the approach.

Weaknesses:

1. Dividing images into a few regions to interpret the decision-making is simplistic. The proposed approach sum the pixel values in a region which may squash important variance cues in a region. Additionally, a coarse grid such as 4X4 or 8X8 may not respect object boundaries. What are the authors' comments on these issues?

2. Authors claim to capture interactions between regions which is missing in the existing works. However, the way the interaction is measured is effectively based on the individual importance of the regions. I am missing what aspect of Eq 11 captures non-linear region dependencies for decision-making that are missing in Eq 9?

3. Authors mentioned the G^{-1} and H^{-1} functions. Where are these inverse functions used?

4. Can G and H be combined in a single encoder and G^{-1} and H^{-1} combined in a single decoder? For example, one can combine G and H in a single encoder such as VAE and consider the latent representation as the lower dimensional representation $r$. Is the proposed approach applicable to such representations?

5. As in figure 3 (a), the interactions between the regions increase with a lower number of regions. This is counterintuitive as discussed in section 4.2.3. This raises the question of how the approach depends on the size of the regions. It will also be useful to visually inspect the results on a set of images while changing the size of the regions.

**Summary Of The Paper:**

This paper presents a sample-based approach for interpreting the decision-making of neural networks for an image classification task. An image is divided into multiple regions and the importance of a region is evaluated separately along with its interaction with other regions. A Sobol variance-based sensitivity metric is used to estimate the importance of regions by varying the region and noting the effect on the prediction. The approach is validated with Imagenet validation images with respect to commonly used image classification models such as AlexNet, CGG16, ResNet50, and DenseNet161.

**Summary Of The Review:**

The paper provides an interesting idea of sample-based interpretability estimation. However, there are many aspects that need to be addressed to justify the impact of the approach. Please address the issues in the weaknesses section and I will be happy to reconsider the decision.

---

> ### Author Response · Authors · 2022-11-10
> **RE: Dividing Images into Regions,  the Summation Decision, Interaction, Inverse Function Issues**
>
> We would like to thank the reviewer for your comments and for pointing out sections of the paper that are unclear and require more clarification and explanation. We would also like to address the reviewers points one by one. Our current revision paper to respond to the reviews is a bit long but we will pair it down. We wanted to start the dialogue in the main time.
>
>
> ### The choice of regions
> We selected regions because the only prior work of which the Author's are aware, which compare the extent to which CNN models use interactions used regions when they did this exploration.
>
> - Wieland Brendel and Matthias Bethge. Approximating cnns with bag-of-local-features modelsworks surprisingly well on imagenet. volume 7, 2019.
>
> The authors used different levels of grids to demonstrate that our method can accurately select the most and least important regions regardless of scale so that future results can be trusted. That the regions do not respect boundaries is not necessarily bad. It depends on the users purpose. To understand the general behavior of models without focusing on particular objects, this is fine. However, to perform direct measurement on the importance of measurements this could be an issue.
>
> *We note that our method is generalizable to any area in an image. To demonstrate this, we have implemented our pipeline for segmentations determined by ImageNet S19 which are human annotated and respect strict boundaries between objects and the background as well as Salient ImageNet which are machine annotated segments that distinguish between the area of an image that should be relevant to predictions and those that are not. These areas do not respect strict boundaries.*
>
> - Shanghua Gao, Zhong-Yu Li, Ming-Hsuan Yang, Ming-Ming Cheng, Junwei Han, and Philip H. S.
> Torr. Large-scale unsupervised semantic segmentation. CoRR, abs/2106.03149, 2021.
>
> - Sahil Singla and Soheil Feizi. Causal imagenet: How to discover spurious features in deep learning?
> CoRR, abs/2110.04301, 2021.
>
> *The green text shows our updated results, as do Figures 4, 5, and 6. To summarize, with these segments we were additionally able to, through direct measurement, show that modern models rely more on the background when making decisions than older models, and that robust models reduce this reliance*
>
> ### Summation
> We would say that, given the that our mapping is such that we can reconstruct the original image regions based on the sum of the pixel values in the region, we actually won't lose variance information during our process. Our process, in more verbal detail is as follows. *In the paper, we have illustrated this pipeline in Figure 1 and have added text (red) that speaks about this in more detail and adds two considerations we had when choosing G and H.*
>
> 1. Select relevant regions using segmentation, region patches, etc...
> 2. Sum the pixel intensities of these regions and construct a vector from this
> 3. Create samples for the Sample Based sensitivity method (Sobol) that is uniform between some bounds centered on the original vector constructed from the regions
> 4. Implement the first part of the inverse mapping which consists of normalizing the regions obtained by the original sum of the pixels in that region and multiplying it by the samples
> 5. Replace the original values of each region with the values of these sampled regions
> 6. The result are image samples that amplified and masked selected regions to varying degrees which will give the Sobol method a wealth of information about how the model responds to changes in the user selected regions of interest.
> 7. Evaluate the model on these samples and pass these outputs to the Sobol method. Since the sampling was done in a vector space that dictated how the regions were modified, the analysis is with respect to the regions
>
> ### Interaction
> *The authors have augmented equation 9, which is now 13, with equations 11 and 12. We also have added more description for the equation in blue text.* The total index of a particular feature can be effectively written as the summation of the primary index of that feature and every higher order feature involving that variable. The ratio between the primary feature and the total index, as a result, says the amount of the total index that is accounted for by the importance of the feature itself and we can imply from that the relative importance of the interactions.
>
> ### Where are $G^{-1}$ and $H^{-1}$ used?
> The authors used functional notation for $G^{-1}$  and $H^{-1}$. However, they are more accurately represented
> as any mechanism that maps from the reduced variable representation back to the original input
> space. *This has been clarified in the paper in orange text and we described the inverse process in Figure 1. In the above enumeration of our pipeline $H^{-1}$  is 4 and $G^{-1}$is 5.*

---

> ### Author Response · Authors · 2022-11-10
> **RE: Autoecoder and Region Size Interaction**
>
> ### Can this be used with an Autoencoder?
> The proposed method is applicable to this situation. However, some considerations need to be made depending on the goal. If the goal is to understand the sensitivity of the decoder to some reduced dimensional representation of an original input then the proposed approach can be applied without modification. However, if the goal is to understand how a model that takes the output of the decoder as an input behaves with respect to the latent representation, some tests should be done to see how sensitive the decoder is to different components of the latent representation so that the user can discern the difference between the model's sensitivity to latent space and the decoders.
>
> Additionally, as we have now added to the paper, an ideal inverse mapping should be such that samples which yield changes in the reduced dimensional representation of an image should yield changes in the generated image. Sample based sensitivity methods like Sobol generate the optimal samples in vector space to allow them to explore how the model output changes due to changes in different components of the input. If a reverse mapping does not propagate these changes to the original input then it is effectively reducing the information these methods obtain for a given sample size.
>
> Avoiding these complexities for a paper primarily about demonstrating the utility of our pipeline is one of the reasons we chose to start with a a simple summation metric that we found to be experimentally effective.
>
> ### Region size examples
> We agree that this will be helpful. The authors will add this to the appendix.

---

> ### Author Response · Authors · 2022-11-17
> **Revision Addressing Comments**
>
> We have updated the revisions we discussed in our response to the reviewer's comments to fit into the 9 page limit. The current revision is color coded so that the reviewers can see what was changed more easily.
>
>  To review.
>
> Red: Clarifies $G$ and $H$, and makes our math more precise, as well as gives an image illustrating the specific pipeline.
>
> Blue: Clarifies how Eq 11 (now Eq 13)  quantifies interaction
>
> Green: Clarifies why we chose regions (to compare to prior results about region interaction) as well as discusses additional results using regions determined by two types of segmentation.
>
> The appendix has an image that shows the impact of shrinking region sizes.
>
> We would like to get feedback from the reviewer on our response to the original reviewer, as well as the revision.

---

> ### Author Response · Authors · 2022-12-14
> **Request for Post Rebuttal Review**
>
> Dear Reviewer phNs,
>
> The discussion window is coming to a close. We would like to ask for a response to our rebuttal version and the responses we gave to address your above concerns.
>
> Best,
>
> Authors

---

### Official Review · Reviewer_yYjF · 2022-10-28

**Confidence:** 3
**Correctness:** 2
**Technical Novelty And Significance:** 3
**Empirical Novelty And Significance:** 2
**Recommendation:** 3

**Clarity, Quality, Novelty And Reproducibility:**

Novelty is high, and I am always in favour of not re-invent the wheel. Sobol is worth further study. However, the quality of the paper in terms of clarity is only average. Reproducibility is limited since no information can be found on which images of ImageNet have been tested.

**Strength And Weaknesses:**

I like first, the questions asked and second the idea of exploiting known methods from feature relevance analysis to understand the output of black-box models. I am kind of disappointed about the implementation of this general, very promising idea. The details of combining Sobol with SBSA are insufficient to support their claims. For example, why shall the sum of pixels in a region be a proper semantic representation of an image(part)? Also, the generation of the low-dimensional feature is arbitrary by the selection of image patches. I am missing a detailed study and motivation for this step. Second, experimental results lack clarity and insight. When comparing with other methods to explain the output of a CNN, their metric is not always the best. Overall, the metrics used, for example, PIR, deliver interesting insight, but in my opinion not more. Figure 3 shows an observation for a very specific pipeline of SBSA and Sobol, and not even one that I find reasonable (see my comment on a feature that uses the sum of intensity values). In summary, the idea is very promising but the current result is premature and without a significant contribution to the area.

**Summary Of The Paper:**

The authors suggest a new method for interpreting decisions by deep learning models using a variance-based sensitivity analysis method. By such analysis of the input features the authors claim that they can analyse not only the contribution of individual features but their relation as well, for example, such image patches. The method is evaluated in an experimental setup using image net and some known CNNs.

**Summary Of The Review:**

This paper is not ready for publication. More needs to be done to arrive at significant statements about the development of CNN architectures and their impact on the representation capability in the model. Compared to other works on interpreting CNN output, this paper lacks theoretical contribution and experimental evaluation.

---

> ### Author Response · Authors · 2022-11-10
> **Choice of Pipeline and Clarity of Experimental Results**
>
> We would like to thank the reviewer  for their comments and for pointing out aspects of the paper that need clarification and will address the comments. We note that our current revision paper is a bit over length because of things we added, but we will shrink it. However, we wanted to start the conversation.
>
> ### Why choose image patches?
> The choice of image regions was selected to be consistent with work done by Brendel et al when they indirectly explored the evolution of the importance of interactions to CNN models.
> -Wieland Brendel and Matthias Bethge. Approximating cnns with bag-of-local-features modelsworks surprisingly well on imagenet. volume 7, 2019.
>
> Since we compared our conclusions  with theirs, we have confidence in it, and the pipeline that yielded it. However, *the authors have added results based on regions determined by segmentations as well (Salient ImageNet which segments the important parts of the image vs the background without but does not necessarily respect boundaries of objects since it is machine annotated and Imagenet-S 919 which are human annotated segments)*
> - Shanghua Gao, Zhong-Yu Li, Ming-Hsuan Yang, Ming-Ming Cheng, Junwei Han, and Philip H. S. Torr. Large-scale unsupervised semantic segmentation. 2021.
>
> -Sahil Singla and Soheil Feizi. Causal imagenet: How to discover spurious features in deep learning? 2021.
>
> *The results of this are shown in the green text and the updated results and measurements are in Figure 4, 5, and 6.*
>
>
> ### Details of Combining Sobol with SBSA
> We would like to clarify that the sum of pixels in a region is not a semantic representation of an image part. The semantic representation here are image parts which are semantic representations of the image. The summation of the pixels are used to form vectors that are sampled in such a way that they determine the extent to which the regions they are associated with are masked and amplified in order to produce sample images that then fed into the relevant model so that the outputs can be analyzed with the Sobol method. *We have added Figure 1 to clarify their pipeline as well as added more descriptions in the text (in red) to show what makes a good mapping and describe our methodology in more detail.* In summary, our pipeline is as follows:
>
> 1. Select relevant regions using segmentation, region patches, etc...
> 2. Sum the pixel intensities of these regions and construct a vector from this
> 3. Create samples for the Sample Based sensitivity method (Sobol) that is uniform between some bounds centered on the original vector constructed from the regions
> 4. Implement the first part of the inverse mapping which consists of normalizing the regions obtained by the original sum of the pixels in that region and multiplying it by the samples
> 5. Replace the original values of each region with the values of these sampled regions
> 6. The result are image samples that amplified and masked selected regions to varying degrees which will give the Sobol method a wealth of information about how the model responds to changes in the user selected regions of interest.
> 7. Evaluate the model on these samples and pass these outputs to the Sobol method. Since the sampling was done in a vector space that dictated how the regions were modified, the analysis is with respect to the regions
>
> ### The Clarity and insights of the Experimental Results and our models not always being the best
> We would like clarification of this point. Both Figures 2 and 3 (1 and 2 originally) which make comparisons to other methods show that our method is the best. Can the reviewer clarify by what they mean by stating that this is not always the case?
>
> Figure 2 plots the change in ground truth score when the $n$ most important regions are masked vs when the $n$ least important regions are. We ran a random baseline (which in this plot is essentially diagonal). The ideal situation is when the bottom $n$ regions have no importance at all. In this case, the slope on this plot would be vertical, which the author's have also plotted. The regions determined by our total index metric from Sobol is the closest to the ideal in all three region sizes used.
>
> Figure 3 plots the sensitivity-n correlation metric which randomly masks $n$ regions of an image and uses the correlation between the correlation between the change in model output between and the sum of the attribution methods of these $n$ regions to determine the effectiveness of sensitivity methods. Here we again run a random baseline, and provide the ideal situation, which no model can hit (which is the correlation being 1). Our metric, $S_{T_i}$ is also the most ideal here. In both of the plots our other metric $S_i$ is not always the best. However, as stated in the paper, $S_i$ is a measurement of how important features are individually where as  $S_{T_i}$ measures how important they are individually and due to their interactions with all other features. It would not be expected for $S_i$ to perform similarly.

---

> ### Author Response · Authors · 2022-11-10
> **RE: Contributions of the Paper**
>
> We will address each point. In the field of interpretability additional insight, especially insight based on the interactions of features is highly novel. Within this field the authors are only aware of a few works applied to deep learning and they all require pairwise comparisons. The $PIR$ gives information about a feature's interaction with every other feature without needing extra calculations since the Saltelli method that the authors use calculate $S_i$ and $S{T_i}$ in one pass.
>
> - Joseph D. Janizek, Pascal Sturmfels, and Su-In Lee. Explaining explanations: Axiomatic feature
> interactions for deep networks, 2020
>
> - Hao Zhang, Yichen Xie, Longjie Zheng, Die Zhang, and Quanshi Zhang. Interpreting multivariate
> interactions in dnns. 2020
>
> Additionally, the pipeline we have proposed provides a framework for performing interpretability at semantic levels which is highly novel within the field. We then performed a case study on a series of real models to explore different questions about these models through direct measurement and were able to reach explore characteristics about the interaction of regions through direct measurement (which to our knowledge has only be done indirectly once).
>
> *To show the generalizability of our pipeline and further reinforce our results, the author's have added experiments in which Sobol-SBSA was applied to two different types of segmentations. ImageNet-S which is a human annotated segmentation set that strictly respects object boundaries and Salient ImageNet which is machine annotated and segments out the core important regions of an image with respect to the background. These do not strictly respect object boundaries. The additional results are described in the green text and we show through direct measurement by the Sobol Indices (calculating the difference between $S_{T,segment} - S_{T, backgronds}$) that the trend in more modern CNN models was to rely more on the background and that robust models reduced this impact. These are both things that, to our knowledge, have not been directly measured (only implied) and demonstrates the ability of our pipeline to go beyond measuring importance of pixels and individual words.*
>
> We would like more clarity on why these are not viewed as significant contributions to the field of interpretability by the reviewer?
>
> ### Pipeline Choice
> The reviewer states that they do not find our pipeline reasonable. With respect to our pipeline, as we described, it essentially create samples images that amplify and mask different areas of the image in such a way that the different sampling based sensitivity methods can obtain a rich set of information about how a CNN output changes with respect to changes in the image and can effectively rank the importance of these areas. The sampling is done based on the method usually done by each of sensitivity analysis method. Moreover, we validated our pipeline by confirming that it is effective at selecting the most and least important regions of an image to a CNN model regardless of scale. Why is this, in the reviewer's view, unreasonable?
>
> ### Comparisons to other CNN Works
>  The reviewer's statement that this paper lacks theoretical contribution and experimental evaluation is abstract. We note that this paper introduces a novel black box method for interpretability which allows direct measurement of the importance and interaction of semantic regions. We used CNNs and ImageNet as a case study for this and have done the following experiments:
> 1. Compared our Sobol pipeline with other pipelines using two metrics for measuring the quality of attribution and found it to perform well
> 2. Directly measured how CNNs utilize interaction between regions. Initially in region patches, but *now also with two forms of segmentation. (Figure 4a)* Prior interpretability methods and pipelines have not provided the ability to perform such direct measurements and we are only aware of one paper that has attempted to measure it, and they were only able to do so indirectly. (Wieland Brendel and Matthias Bethge. Approximating cnns with bag-of-local-features models works surprisingly well on imagenet. volume 7, 2019.)
> 3. *We have now additionally used the application of Sobol-SBSA to directly measured the difference between the importance of segmented regions and the backgrounds to varying CNN models and saw that this shrunk (the dependence on background grew) as CNNs evolved (Figure 4b)* This is also another quantity that to our knowledge has not been directly measured.
> 4. *We added the calculations of 2 and 3 for robust vs normal versions of CNNs and saw that robust models reduced the reliance of CNNs on background and interactions*
>
> If these experiments are not enough to demonstrate how our pipeline can be used to directly answer questions about models that other interpretability pipelines don't, would the reviewer have specific examples of other work, and the type of experimental and theoretical contribution that is missing?

---

> ### Author Response · Authors · 2022-11-10
> **Reproducability**
>
> ### Reproducability
>
> It is not common place for specific images to be specified. However, since the authors explicitly stated that the sets of data used were Validation ImageNet, Validation Segmentation ImageNet-S 919, and Salient ImageNet training, and since the experiments were performed with statistically significant number of samples (1000 for comparisons to other methods and 10000 for interaction and background subtraction experiments), reproducibility is strong.

---

> ### Author Response · Authors · 2022-11-17
> **Revision Addressing Comments**
>
> We have updated the revisions we discussed in our response to the reviewer's comments to fit into the 9 page limit. The current revision is color coded so that the reviewers can see what was changed more easily.
>
> To review.
>
> Red: Clarifies  and , and makes our math more precise, as well as gives an image illustrating the specific pipeline.
>
> Blue: Clarifies how Eq 11 (now Eq 13) quantifies interaction
>
> Green: Clarifies why we chose regions (to compare to prior results about region interaction) as well as discusses additional results using regions determined by two types of segmentation.
>
> We would like feedback from the reviewer on our original rebuttal as well as the revisions.

---

> ### Author Response · Authors · 2022-12-14
> **Post Rebuttal Review**
>
> Dear Reviewer yYjF,
>
> The discussion window is coming to a close. We would like to ask for a response to our rebuttal version and the responses we gave to address your above concerns.
>
> Best,
>
> Authors

---

### Official Review · Reviewer_1XwS · 2022-10-31

**Confidence:** 3
**Correctness:** 3
**Technical Novelty And Significance:** 2
**Empirical Novelty And Significance:** 2
**Recommendation:** 3

**Clarity, Quality, Novelty And Reproducibility:**

The method is fairly simple and most of them are from the previous works, so I believe it should be easy to reproduce. On the other hand, the novelty is a concern as mentioned above.

**Strength And Weaknesses:**

- The main criticism is on the novelty. Most of the components have been used in Fel et al., 2021.   Until equation (11), most of the contents are covered by Fel et al 2021, and the new ingredients from this paper seems to be equation (12) only.  I would encourage authors to spend more space discussing the new components proposed by this paper instead of repeating what people have done before. Also, It would also be nice to provide literatures about Sobol methods,  as Fel et al., 2021 is not the one proposing this metric but proposing to reuse this tool. Some important literatures include
    * I. Sobol’, “Sensitivity estimates for non linear mathematical models,” Mathematical Modelling and Computational Experiments, vol. 1, pp.
407–414, 1993.
    * B. Li and C. Chen, “First-order sensitivity analysis for hidden neuron selection in layer-wise training of networks,” Neural Processing Letters, vol. 48, pp. 1105–1121, 2018.

- It is not clear to me how we construct the inverse function of H and G.  The inputs of H and G are with higher dimensions than their outputs, which the inverse functions "usually" don't exist. Also, the function includes some pooling operations (sum), it is counter-intuitive to be able to construct an exact inverse function. More descriptions would be helpful for readers to understand.

- The proposed methods are based on comparing the variances of different components, such as equation (10) and equation (11). However, there are already a lot of literatures discussing the variance of features for interpretability. The oldest one might from linear regression literatures, such as the notion of R^2. There are also more recent one worths discussing in the literature.

- Other minor points:
    - The authors criticize the limitation of GradCAM is heavily depending on the architecture, which I think it's not a proper statement, as it has been widely applied to many different architectures with practical success. I would like to know authors' opinion on it.

**Summary Of The Paper:**

The authors proposed an algorithm for interpreting deep models, which is an extension of Fel et al., 2021. The main tool it relies on is called Sobol first-order sensitivity. The experiments are demonstrated on different CNN variants, including VGG, ResNet and DenseNet.

**Summary Of The Review:**

The main concern is the novelty. There are also some components which seems not right to me, and I would like to see authors' justification in the rebuttal.

---

> ### Author Response · Authors · 2022-11-10
> **Addressing the Novelty**
>
> We would like to thank the reviewer for their comments on the paper and pointing out aspects of it that require clarification. We will address the points made in the review here. Our paper is currently a little long because of some things that we added, but we wanted to move forward with the conversation. We will shorten it as we go.
>
> ### Novelty
> We do not agree with the statement that we are only redoing what Fel et al did. Fel et al focused primarily the use of Sobol as a measure of importance. In the field of interpretability within Deep Learning, measuring the interactions of features is very new. For all the methods of which we are aware, pairwise comparisons of features must be calculated and it must be done separately from quantifying the importance of individual variables.
>
> - Joseph D. Janizek, Pascal Sturmfels, and Su-In Lee. Explaining explanations: Axiomatic feature
> interactions for deep networks, 2020
> - Hao Zhang, Yichen Xie, Longjie Zheng, Die Zhang, and Quanshi Zhang. Interpreting multivariate
> interactions in dnns. 2020.
>
> Equation (11) which is now (13) provides a measure of the extent to which a feature interacts with all other features without needing to do an extra calculation since Sobol obtains both the primary and total index in one calculation. The insight one can get from this is highly novel within Deep Learning and Fel et al's paper did not explore this.
>
> In addition to this, however, this paper has several other contributions that Fel et al do not. Fel et al focused on the use of Sobol only as another tool for measuring importance.
>
> 1. We provide a framework for performing interpretability at a semantic level (regions and segments when it comes to images). *We added some ideal properties for G and H in these functions (red) as well as illustrated visually our specific pipeline to clarify the process (Figure 1)*
> 2. We performed a case study in which we used metrics related to Sobol and our framework to directly measure aspects of how CNNs use input regions in a way that has not been done. We did this only for regions in the original paper, *but added the application of our pipeline to two sets of segmentation data to show its generalizability. ImageNet-S which respects boundaries and is human annotated and Salient ImageNet which is a machine annotated ImageNet dataset that segments the area of an image that should be relevant to predictions vs those that should not be. These segments do not respect boundaries [1,2].*
> 3.  Based on these we were able to through direct measurement show that more modern CNNs rely more on background information and on interaction on regions. We were also able to show through direct measurement that robust versions of modern CNNs mitigate this impact. *Figure 4, 5, and 6 in the revision and the green text*  To our knowledge, *our pipeline is unique in being able to directly quantify these properties about the importance of semantic components of inputs to model outputs (region patches and segments in this case). The things we have measured here, to our knowledge, have only been indirectly explored (e.g. by adding noise to be background of images) [3,4]*
>
> ### Literature about Sobol
> We are aware that Fel et al did not propose the metric. We cited the seminal work of Saltelli while describing the Sobol method [5]. This paper is the foundation of modern sampling based implementations of Sobol.  We additionally cited a popular open source Sensitivity analysis library that we used for implementing Sobol [6]
>
> [1] Shanghua Gao, Zhong-Yu Li, Ming-Hsuan Yang, Ming-Ming Cheng, Junwei Han, and Philip H. S. Torr. Large-scale unsupervised semantic segmentation. CoRR, abs/2106.03149, 2021.
>
> [2] Sahil Singla and Soheil Feizi. Causal imagenet: How to discover spurious features in deep learning? CoRR, abs/2110.04301, 2021.
>
> [3] Wieland Brendel and Matthias Bethge. Approximating cnns with bag-of-local-features modelsworks surprisingly well on imagenet. volume 7, 2019.
>
> [4] Sahil Singla, Mazda Moayeri, and Soheil Feizi. Core risk minimization using salient imagenet,
> 2022
>
> [5] Andrea Saltelli, Paola Annoni, Ivano Azzini, Francesca Campolongo, Marco Ratto, and Stefano
> Tarantola. Variance based sensitivity analysis of model output. design and estimator for the total
> sensitivity index. Computer Physics Communications, 181, 2010
>
> [6] on Herman and Will Usher. Salib: An open-source python library for sensitivity analysis. Journal
> of Open Source Software, 2, 2017

---

> ### Author Response · Authors · 2022-11-10
> **H and G, other Variance Methods, Grad CAM**
>
> ### Clarifying H and G
> *We have added Figure 1 to clarify our pipeline as well as more description in red in the most recent submission. We additionally added explicitly the two properties that we considered when choosing $G$ and $H$*  Although we used functional notation for simplicity, our pipeline simply meant these as any mechanisms that facilitate the forward and backward transformations between $x$ and $r$. We also describe our pipeline below for further clarification. To summarize the pooling operator on selected regions are used to construct a vector from which the sensitivity method of interest constructs samples (uniform samples with the original values as the mean). The selected regions, in the backwards pass, are normalized by the original sum and multiplied by the sample. The result is the generation of samples of images that have regions which are masked and amplified to varying degrees and which give the Sobol method (or any other sample based sensitivity method) information with which to discover the impact of the importance of regions. We describe the exact steps below. The inverse functions occur at steps 4 and 5.
>
> 1. Select relevant regions using segmentation, region patches, etc...
> 2. Sum the pixel intensities of these regions and construct a vector from this
> 3. Create samples for the Sample Based sensitivity method (Sobol) that is uniform between some bounds centered on the original vector constructed from the regions
> 4. Implement the first part of the inverse mapping which consists of normalizing the regions obtained by the original sum of the pixels in that region and multiplying it by the samples
> 5. Replace the original values of each region with the values of these sampled regions
> 6. The result are image samples that amplified and masked selected regions to varying degrees
> 7. Evaluate the model on these samples and pass these outputs to the Sobol method. Since the sampling was done in a vector space that dictated how the regions were modified, the analysis is with respect to the regions
>
> ### Comparison to other variance methods
>
> We start by discussing the idea of the literature that already exists. Multiple aspects of major black box methods for understanding Deep Learning models already existed prior to Deep Learning, if not the exact method. Shapley Values and Integrated Gradients are great examples of this [1,2]
>
> However, Deep Learning models are uniquely challenging because of the unprecedentedly non-linear nature of the functions and the sheer size of the inputs. This is where our work falls into.
>
> We do not agree with equating the older methods of variance study in literature with Sobol. Variance measures such as $R^2$ capture the extent to which a linear regression fits data, but they do not give any explicit measurements about how the input variables interact. For example, a linear regression would give a low $R^2$ score for both $y = cos(x_1) + sin(x_2)$ and $y = x_1sin(x_2)$ whereas Sobol would be able to distinguish that in the first equation x1 and x2  are important to y individually, but that they interact in the second. Additionally, to our knowledge, other methods of calculating the importance of variance components with respect to interactions either make assumptions about the functional form of the data (Multi-variate linear regressions, etc...) or require pairwise comparisons. Sobol does not require any of these which makes it uniquely suited for using its results to understand how Deep models utilize interactions and which is why we chose it for our pipeline for understanding how models use some semantic representations of inputs.
>
> Would the reviewer have a specific example of other work that explores the use of Sobol variance to understand interactions in Deep Learning?
>
> ### GradCam
> GradCam is useful in many applications. However, it has limited ability to understand model behavior holistically. This is because it was designed to be able to localize certain objects relevant to an output while backpropping through architectures. One of the common axioms needed for an ideal sensitivity method is the following. If perturbing part of an input perturbs the output then a good attribution method should assign it non-zero attribution. Since GradCam uses RELU to ignore negative signals that don't contribute positively to a particular classification output, it is possible for it to give zero attributions to parts of an image that should not have zero attribution simply because they act as evidence against a particular output. Unlike our Sobol method, GradCam will not generally be able to accurately discern the least important areas of an input.
>
> [1]Mukund Sundararajan, Ankur Taly, and Qiqi Yan. Axiomatic attribution for deep networks. Inter-
> national Conference on Machine Learning, 34, 2017.)
>
> [2]I. Kononenko, E. ˇStrumbelj, Z. Bosnic, Darko Pevec, M. Kukar, and M. Robnik-Sikonja. Explana-tion and reliability of individual predictions. Informatica, 37, 2013.

---

> ### Author Response · Authors · 2022-11-17
> **Revision Addressing Comments**
>
> We have updated the revisions we discussed in our response to the reviewer's comments to fit into the 9 page limit. The current revision is color coded so that the reviewers can see what was changed more easily.
>
> To review.
>
> Red: Clarifies  and , and makes our math more precise, as well as gives an image illustrating the specific pipeline.
>
> Blue: Clarifies how Eq 11 (now Eq 13) quantifies interaction
>
> Green: Clarifies why we chose regions (to compare to prior results about region interaction) as well as discusses additional results using regions determined by two types of segmentation.
>
> We would like feedback from the reviewer on our original rebuttal as well as the revisions.

---

> ### Author Response · Authors · 2022-12-14
> **Request for Post Rebuttal Review**
>
> Dear Reviewer 1XwS,
>
> The discussion window is coming to a close. We would like to ask for a response to our rebuttal version and the responses we gave to address your above concerns.
>
> Best,
>
> Authors

---

### Author Response · Authors · 2022-11-18
**Author's Summary**

We would like to thank the reviewers for their valuable feedback in pointing out the things that were unclear about the paper, the need to better illustrate our methodology and contributions, and the usefulness of showing our pipeline for more than one type of image region. We haven't gotten feedback to our rebuttals, but we responded in detail to the comments of each reviewer below and addressed their comments within our manuscript.

### Changes to Manuscript
We summarize the changes to the manuscript below.
- Reduced the relevant work section to create room for us to discuss our methodology and results in more detail (R1)
- Expanded on the discussion of our framework for performing sensitivity analysis of semantic features of inputs, added an illustration, and modified our nomenclature to be more mathematically accurate (R1,R2,R3)
- Added considerations for what makes for ideal $G$, $H$, and $R$ functions in our framework
- Clarified how $PIR$ is a measure of interaction by showing its derivation (R3)
- Clarified that the reason we initially picked image regions as our semantic feature was to make direct comparisons to the results of [1]. (R1, R2, R3)
- Added implementations of our pipeline for segmentations determined by two different types of work (one that respects object boundaries and one that does not) [2,3] (R2)
- Differentiated more explicitly between results that validated the reliability of our pipeline by correctly replicating other results and those that were new (R1, R2, R3)
- Added to the appendix plots that demonstrate how our pipeline can have issues when region sizes become very small (R3)

[1] Wieland Brendel and Matthias Bethge. Approximating cnns with bag-of-local-features modelsworks surprisingly well on imagenet. volume 7, 2019.

[2] Shanghua Gao, Zhong-Yu Li, Ming-Hsuan Yang, Ming-Ming Cheng, Junwei Han, and Philip H. S. Torr. Large-scale unsupervised semantic segmentation. CoRR, abs/2106.03149, 2021.

[3] Sahil Singla and Soheil Feizi. Causal imagenet: How to discover spurious features in deep learning? CoRR, abs/2110.04301, 2021.

---

### Decision · Program_Chairs · 2023-01-20

**Decision:**

Reject

**Justification For Why Not Higher Score:**

Because the technical/empirical novelty and significance of the paper is limited

**Justification For Why Not Lower Score:**

N/A

**Metareview: Summary, Strengths And Weaknesses:**

This paper proposes a sample-based approach for interpreting neural net decisions. I find the main findings of the paper about region interactions and reliance on the background interesting. However, the agreement among the reviewers is that the technical/empirical novelty and significance of the paper is currently marginally below acceptance standards and is not ready for publication. Therefore, I recommend rejection. I want to recommend authors to take reviewers suggestions into account to improve the paper. In particular, I find reviewer phNs's suggestions to be very helpful for improving the quality of the paper.